# Design and Fabrication of a Novel Wheel-Ring Triaxial Gyroscope

**DOI:** 10.3390/s22249978

**Published:** 2022-12-18

**Authors:** Tianqi Guo, Wenqiang Wei, Qi Cai, Rang Cui, Chong Shen, Huiliang Cao

**Affiliations:** Key Laboratory of Instrumentation Science & Dynamic Measurement, Ministry of Education, North University of China, Taiyuan 030051, China

**Keywords:** micro-electro-mechanical system (MEMS) gyroscope, motion equation, capacitance formula, operating mode, resonant ring

## Abstract

This paper presents a new type of three-axis gyroscope. The gyroscope comprises two independent parts, which are nested to further reduce the structure volume. The capacitive drive was adopted. The motion equation, capacitance design, and spring design of a three-axis gyroscope were introduced, and the corresponding formulas were derived. Furthermore, the X/Y driving frequency of the gyroscope was 5954.8 Hz, the Y-axis detection frequency was 5774.5 Hz, and the X-axis detection frequency was 6030.5 Hz, as determined by the finite element simulation method. The Z-axis driving frequency was 10,728 Hz, and the Z-axis sensing frequency was 10,725 Hz. The MEMS gyroscope’s Z-axis driving mode and the sensing mode’s frequency were slightly mismatched, so the gyroscope demonstrated a larger bandwidth and higher Z-axis mechanical sensitivity. In addition, the structure also has good Z-axis impact resistance. The transient impact simulation of the gyroscope structure showed that the maximum stress of the sensitive structure under the impact of 10,000 g@5 ms was 300.18 Mpa. The gyroscope was produced by etching silicon wafers in DRIE mode to obtain a high aspect ratio structure, tightly connected to the glass substrate by silicon/glass anode bonding technology.

## 1. Introduction

A gyroscope is a sensor that measures the rotation speed of objects [1,2,3]. In his experiment on the Earth’s rotation, Jean Foucault found that a swing pendulum on a rotating platform would slowly rotate in a clockwise direction, with the tip of the swing changing constantly. He named the instrument a gyroscope, the Greek words gyro (to spin) and skopeein (to look). Early gyroscopes were generally large in volume and high in cost. The appearance of the MEMS system promoted the formation of micromechanical gyroscopes. Therefore, MEMS gyroscopes have the advantages of being easy to carry and having low power consumption [4,5].

In the continuous development of micromachined gyroscopes, various new forms of gyroscope structures have emerged, such as tuning fork gyroscopes, ring gyroscopes, laser gyroscopes, and hemispherical gyroscopes. The number of angular rates detected by the gyroscope also shifts from single to multi-axis angular rates.

In practical application, it is often necessary to obtain the angular rate information of the three axial angles of the carrier: pitch angle, yaw angle, and roll angle. Therefore, developing a new type of three-axis gyro structure has become a trend. With the development of MEMS processing technology and gyroscope design technology, many institutions have conducted in-depth research on monolithic three-axis micromechanical gyroscopes, such as a monolithic three-axis MEMS gyroscope structure with eight symmetrical plates developed by the University of Pisa [6]. However, some three-axis gyroscope structures are complex, difficult to process, and have other issues, and Z-axis anti-high overload capacity is unknown, as shown in Table 1. This paper proposes a new three-axis gyroscope to provide ideas for future gyroscope design. The design has a tiny size (8.0 mm × 8.0 mm), simple structure, excellent Z-axis resistance to overload, and high z-axis sensitivity. The two mutually independent parts also avoid coupling problems. In this paper, the structure is firstly analyzed in detail, and the corresponding theoretical equations are derived. Then the structure is further simulated to verify its sensitivity and impact resistance. Finally, the detailed fabrication is introduced, and the finished product diagram is given after processing.

## 2. Structure Design

### 2.1. Operating Principle

A schematic diagram of the three-axis gyroscope structure introduced in this paper is shown in Figure 1. The gyroscope consists of two independent parts: the external structure includes an outer resonant ring and eight S-shaped springs for support; the internal design consists of a Y-axis sensing frame (M_1_, an inner cylindrical frame, X-Y plane out of motion, and a torsion around the X-axis; used to sense the Y input angular rate), Y-driven flexible joint, wheel structure driven frame (M_2_, a medium cylindrical shell frame, connected with four symmetrical anchors by four support beams; used to support the whole frame), X-driven flexible joint, and X-axis sensing frame (M_3_, an outer cylindrical frame, X-Y plane out of motion, torsion around the Y-axis; used to sense the X-axis input angular rate).

The driving flexible joint transmits the electrostatic force generated by the driving comb to drive the X and Y-axes’ sensing masses to rotate around the Z axis. When there is an angular rate input in the X axis, the sensing mass M_3_ will twist around the Y axis under the influence of the Coriolis force. Similarly, the Y-sensing mass M_1_ will twist around the X axis when the Y axis has an angular rate input. The distance between the X/Y sensing mass and the bottom electrode changes after the torsion pendulum around the axis, and the angular input rate in the X/Y direction can be obtained by detecting the change of capacitance.

### 2.2. Mathematical Model

The vibrating gyroscope works through the Coriolis effect. The Coriolis force is proportional to the rotational angular velocity. Therefore, the angular velocity of the rotating system can be obtained by detecting Coriolis force. Because the structure adopts an independent nested structure [11,12,13,14,15], the ring and wheel structures do not affect one another, and the motion equations [16,17,18,19,20] of the two structures are given, respectively:(1)$$\left\{\begin{array}{l} m_{1}\ddot{x}+c_{1}\dot{x}+k_{1}x=f_{drive}\\ m_{2}\ddot{y}+c_{2}\dot{y}+k_{2}y=f_{coriolic} \end{array}\right.$$

Equation (1) is the motion equation of the resonant ring, where *x* and *y* are the amplitudes of the driving and sensing modes, *m*_1_, *m*_2_ is the resonant ring’s effective mass, *c*_1_, *c*_2_ is the damping coefficient, and *k* is the stiffness coefficient [21].
(2)$$\left\{\begin{array}{l} \sum_{i=1}^{3}I_{iz}{\ddot{\theta }}_{z}+\sum_{i=1}^{3}C_{iz}{\dot{\theta }}_{z}+K_{z}^{d}\theta_{z}=F_{d}\\ I_{1y}{\ddot{\phi }}_{y}+C_{1y}{\dot{\phi }}_{y}+K_{1y}^{s}\phi_{y}=2I_{1z}{\dot{\theta }}_{z}\Omega_{x}\\ I_{3x}{\ddot{\phi }}_{x}+C_{3x}{\dot{\phi }}_{x}+K_{3x}^{s}\phi_{x}=2I_{3z}{\dot{\theta }}_{z}\Omega_{y} \end{array}\right.$$

Equation (2) is the motion equation of the wheel structure, where *θ_z_* is the angle at which the wheel structure rotates about the Z-axis; *I_ij_* (*i* = 1, 2, 3; *j* = x, y, z) is the moment of inertia of three parts of the wheel structure, *i* = 1, 2, 3 corresponds successively to the innermost Y-axis sensing mass(M_1_), the wheel drive frame(M_2_), and the outermost X-axis sensing mass(M_3_); *j* = x, y, z corresponds to the three axes (X, Y, and Z); *C_ij_* refers to the damping coefficients; $K_{z}^{d}$ is the stiffness coefficient of the driving mode; *F_d_* is the electrostatic force generated at the comb; $K_{ij}^{S}$ refers to the stiffness coefficients of the sensing mode; Ω*_y_* is the angular rate of the input along the Y-axis. *ϕ_y_* is the displacement of the Y sensing mass rotated around the Y-axis; *ϕ_x_* is the displacement of the X sensing mass revolved around the X-axis. Ω*_x_* is the angular rate of the input along the X-axis. As an example, the X-axis input angular rate change in the angular coordinate system is shown in Figure 2.

For the ring structure n-node bending mode, the point shift change on it can be expressed as [22,23,24,25,26]:(3)$$\left\{\begin{array}{l} u_{r}=nq_{1}(t)\cos n\gamma +nq_{2}\sin n\gamma \\ u_{\gamma }=-q_{1}(t)\sin n\gamma +q_{2}\cos n\gamma \end{array}\right.$$
where u_r_ is the in-plane radial displacement, *u_γ_* is the tangential displacement, *γ* is the angular position, and *q*_1_(*t*) and *q*_2_(*t*) are the generalized coordinate systems.

The ring part of the gyroscope adopts a single-structure resonant ring, as shown in Figure 3. When the resonant ring is working, two-node bending deformation is used, *n* = 2.
(4)$$\left\{\begin{array}{l} u_{r}=2q_{1}(t)\cos2\gamma +2q_{2}\sin2\gamma \\ u_{\gamma }=-q_{1}(t)\sin2\gamma +q_{2}\cos2\gamma \end{array}\right.$$

### 2.3. Stiffness Calculation of S-Shaped Springs

Micro-springs are an essential part of MEMS [27], which can store mechanical energy and provide elastic force. They are usually used in microsensors, micro actuators, and MEMS gyroscopes. Since S-shaped springs can reduce the structural stiffness and capture smaller signals, and the design without a right angle makes the residual stress of these S-shaped springs less than that of the straight beam and the crab beam, combined with the above advantages, S-shaped springs are used in the design of the resonant ring support beam.

The deformation energy of the cantilever mainly includes axial tensile deformation energy and bending deformation energy, so the strain energy and radial displacement of the cantilever are [28,29]:(5)$$\left\{\begin{array}{l} V_{\varepsilon }=\int \frac{F_{N}^{2}\left(x\right)dx}{2EA}+\int \frac{M^{2}\left(x\right)dx}{2EI}\\ \delta =\frac{\partial V_{\varepsilon }}{\partial F} \end{array}\right.$$
where E is the elastic modulus of the S-shaped springs, *M*(*x*) is the bending moment acting on the cross-section of the S-shaped springs, *A* is the cross-sectional area of the S-shaped springs, *F_N_*(*x*) is the positive axial tension acting on the cross-section of the S-shaped springs, and *I* is the moment of inertia of the section of the S-shaped springs. Taking the cantilever beam at this level as an example, the left end of the cantilever beam is fixed at the anchorage point, and a load *F_N_* opposite the X-axis direction is applied to the right end. The simplified structure is shown in Figure 4.

Part ①:(6)$$\begin{array}{ll} \delta_{1} & =\int_{0}^{L_{1}}\frac{F_{N}}{EA}dx\\ & =\frac{F_{N}L_{1}}{EA} \end{array}$$

Part ②:(7)$$M_{2}\left(x\right)=F_{N}R\left(1-\cos\alpha \right)$$

The displacement of Part 2 is:(8)$$\begin{array}{ll} \delta_{2} & =\int_{0}^{\frac{\pi }{2}}\frac{F_{N}R\left(1-\cos\alpha \right)}{EI}R(1-\cos\alpha )d\alpha \\ & =\int_{0}^{\frac{\pi }{2}}\frac{F_{N}R^{2}}{EI}(1-2\cos\alpha +\cos^{2}\alpha )d\alpha \\ & =\frac{F_{N}R^{2}}{EI}\left(\frac{3\pi }{4}-2\right) \end{array}$$

Part ③:(9)$$M_{3}\left(x\right)=F_{N}x,-\left(R+L_{1}\right)\leq x\leq -R$$

The displacement of Part 3 is:(10)$$\begin{array}{ll} \delta_{3} & =\int_{-\left(R+L_{1}\right)}^{-R}\frac{F_{N}x\cdot x}{EI}dx\\ & =\frac{F_{N}\left[-R^{3}+{\left(R+L_{1}\right)}^{3}\right]}{EI} \end{array}$$

Part ④:(11)$$M_{4}\left(x\right)=F_{N}(R+L_{1})+F_{N}\sin\alpha$$

The displacement of Part 4 is:(12)$$\begin{array}{ll} \delta_{4} & =\int_{0}^{\pi }\frac{F_{N}\left(R+L_{1}+\sin\alpha \right)\left(R+L_{1}+\sin\alpha \right)}{EI}d\alpha \\ & =F_{N}\int_{0}^{\pi }\frac{R^{2}+2RL_{1}+2R\sin\alpha +\sin^{2}\alpha }{EI}d\alpha \\ & =\frac{F_{N}\left(2R^{2}\pi +4RL_{1}\pi +2L_{{}_{1}}^{2}\pi +\pi +8R+8L_{1}\right)}{EI} \end{array}$$

Part ⑤:(13)$$M_{5}\left(x\right)=F_{N}x,-\frac{L_{2}}{2}\leq x\leq \frac{L_{2}}{2}$$

The displacement of part 5 is:(14)$$\begin{array}{ll} \delta_{5} & =\int_{-\frac{L_{2}}{2}}^{\frac{L_{2}}{2}}\frac{F_{N}x\cdot x}{EI}dx\\ & =\frac{F_{N}L_{2}^{3}}{12EI} \end{array}$$

Part ⑥:(15)$$M_{6}=\frac{F_{N}L_{2}}{2}+F_{N}R\sin\alpha$$

The displacement of Part 6 is:(16)$$\begin{array}{ll} \delta_{6} & =\int_{0}^{\pi }\frac{F_{N}{\left(\frac{L_{2}}{2}+R\sin\alpha \right)}^{2}}{EI}d\alpha \\ & =\int_{0}^{\pi }\frac{F_{N}(\frac{L_{2}^{2}}{4}+L_{2}R\sin\alpha +\frac{R^{2}-R^{2}\cos2\alpha }{2})}{EI}d\alpha \\ & =\frac{F_{N}\left(L_{2}{}^{2}\pi +2R^{2}\pi +8L_{2}R\right)}{4EI} \end{array}$$

The displacements of Parts 7 and 9 are the same as those of Part 5:(17)$$\delta_{7}=\delta_{9}=\delta_{5}=\frac{F_{N}L_{2}^{3}}{12EI}$$

The displacement of Part 8 is the same as that of the part 4:(18)$$\delta_{8}=\delta_{4}=\frac{F_{N}\left(2R^{2}\pi +4RL_{1}\pi +2L_{{}_{1}}^{2}\pi +\pi +8R+8L_{1}\right)}{EI}$$

The displacement of Part 10 is the same as the that of Part 6:(19)$$\delta_{10}=\delta_{6}=\frac{F_{N}\left(L_{2}{}^{2}\pi +2R^{2}\pi +8L_{2}R\right)}{4EI}$$

The displacement of Part 11 is:(20)$$\begin{array}{ll} \delta_{11} & =\int_{0}^{L_{1}}\frac{F_{N}x\cdot x}{EI}dx\\ & =\frac{F_{N}L_{{}_{1}}^{3}}{3EI} \end{array}$$

The displacement of Part 12 is:(21)$$\begin{array}{ll} \delta_{12} & =\int_{0}^{\frac{\pi }{2}}\frac{F_{N}R\left(1-\cos\alpha \right)R\left(1-\cos\alpha \right)}{EI}d\alpha \\ & =\frac{3\pi F_{N}R^{2}}{4EI}-\frac{2F_{N}R^{2}}{EI} \end{array}$$

The displacement of Part 13 is:(22)$$\begin{array}{c} \delta_{13}=\int_{0}^{L_{3}}\frac{F_{N}}{EA}dx\\ =\frac{F_{N}L_{3}}{EA} \end{array}$$

The formula for calculating the cantilever beam stiffness is:(23)$$K_{B}=\frac{1}{\left[\frac{L_{1}}{EA}+\frac{R^{2}}{EI}(\frac{3\pi }{4}-2)+\ldots +\frac{L_{3}}{EA}\right]}$$

Assuming that the width of the ring is *W_r_*, *m_eff_* is the effective mass of the resonant ring, and *μ* is the Poisson’s ratio, the equation for the stiffness of the resonant ring is as follows:(24)$$K_{r}=\frac{E\pi W_{r}{}^{3}h^{3}}{3R(1-\mu^{2})}$$

Therefore, the resonant frequency can be calculated as:(25)$$\omega_{r}=\sqrt{\frac{K_{r}+K_{B}}{m_{eff}}}$$

### 2.4. Drive Force Calculation

A. Ring Structure

The driving mode of a gyroscope can be divided into electrostatic, piezoelectric, electromagnetic, electric, optical, and so on. Due to the advantages of simple implementation, high sensitivity, and a low-temperature coefficient of electrostatic drive, this method is adopted to excite the gyroscope. The resonant ring adopts a variable spacing drive. The ring part is tested by eight main electrodes and eight auxiliary electrodes. As shown in Figure 5, the pair of 0° and 180° electrodes constitute the driving electrode, the team of 90° and 270° electrodes form the driving detection electrode, the pair of 45° and 225° electrodes comprise the detection electrode, and the pair of 135° and 315° electrodes are the orthogonal electrode. Since the displacement generated by the gyroscope during driving is minimal, the ring capacitor can be equivalent to the plate capacitor, as shown in Figure 6 [30].

The parallel plate capacitance formula can be expressed as:(26)$$C=\frac{\varepsilon \theta Rh}{d}$$
where *ε* is the vacuum dielectric constant, *θ* is the central angle corresponding to the arc, *R* is the radius of the ring, *h* is the structure’s thickness, and *d* is the distance between two plates. When the plate moves Δ*d* and the driving voltage *V_d_* is applied between the two plates, the electrostatic force *F_rd_* is generated as follows:(27)$$\begin{array}{ll} F_{rd} & =-\frac{1}{2}\frac{\partial C}{\partial (\Delta d)}V_{d}^{2}\\ & =\frac{\varepsilon \theta Rh}{2{\left(d+\Delta d\right)}^{2}}V_{d}^{2}\\ & ≅\frac{\varepsilon \theta Rh}{2d^{2}}\left(1+2\frac{\Delta d}{d}\right)V_{d}^{2} \end{array}$$

B. Wheel Structure

The driving part of the wheel structure (M_2_) adopts a scalloped comb and differential electrostatic driving [31]. The driving voltage is *V_wd+_* and *V_wd-_*, respectively:(28)$$\left\{\begin{array}{l} V_{wd+}=V_{D}+V_{A}\sin\omega t\\ V_{wd-}=V_{D}-V_{A}\sin\omega t \end{array}\right.$$

The driving voltage is designed as a superposition of DC (direct current) and AC (alternating current), where *ω* represents the driving mode angular frequency. Therefore, the electrostatic driving torque of the wheel structure can be expressed as:(29)$$\begin{array}{ll} M_{ei} & =\frac{\partial E_{i}}{\partial \alpha }=\frac{\partial \left(\frac{1}{2}C_{wd+}V_{{}_{wd+}}^{2}+\frac{1}{2}C_{wd-}V_{wd-}^{2}\right)}{\partial \alpha }\\ & =\frac{\partial \left(\frac{1}{2}\frac{\varepsilon \left(\alpha +\Delta \alpha \right)}{h_{d}}dr_{i}V_{{}_{wd+}}^{2}+\frac{1}{2}\frac{\varepsilon \left(\alpha -\Delta \alpha \right)}{h_{d}}dr_{i}V_{{}_{wd-}}^{2}\right)}{\partial \alpha }\\ & =\frac{\varepsilon dr_{i}}{2h_{d}}\left(V_{{}_{wd+}}^{2}-V_{{}_{wd-}}^{2}\right) \end{array}$$

In the formula, *C_wd_+* and *C_wd-_* respectively represent the capacitance formed by the movable comb and the fixed comb on both sides, as shown in Figure 7; *α* is the overlapping angle between the movable comb and the fixed comb; Δ*α* is the deflection angle of the movable comb; *h_d_* is the distance between the moving comb and the fixed comb; *r_i_* is the radius of the *i*th moving comb tooth; *h* is the thickness of the structure. After substituting the voltage Formula (28), the electrostatic driving torque is as follows:(30)$$\begin{array}{ll} \sum M_{ei} & =\sum_{i}\frac{\varepsilon hr_{i}}{h_{d}}2V_{D}V_{A}\sin\omega t\\ & =\sum_{i}\alpha \frac{\varepsilon hr_{i}}{h_{d}\alpha }2V_{D}V_{A}\sin\omega t\\ & =\frac{C}{\alpha }2V_{D}V_{A}\sin\omega t \end{array}$$

It can be seen that the electrostatic driving torque is only related to the applied voltage rather than the angular displacement.

### 2.5. Sensing Capacitance

A. Ring Structure

When the resonant ring is subjected to the Coriolis force, the plate spacing changes, and the capacitance changes along with the displacement.

The electrode diagram for the sensing mode is shown in Figure 8. Suppose the driving electrode is at angle *θ*_1_, and the corresponding central angle is *ϕ*; thus, the capacitance *C_rs_* of the sensing electrode can be written as:(31)$$\begin{array}{ll} C_{rs} & =\int_{\theta_{1}-\frac{\phi }{2}}^{\theta_{1}+\frac{\phi }{2}}\frac{\varepsilon Rh}{d}-\frac{\varepsilon Rh}{d-\Delta d}d\theta =\frac{\varepsilon Rh}{d}\int_{\theta_{1}-\frac{\phi }{2}}^{\theta_{1}+\frac{\phi }{2}}1-\frac{1}{1-\frac{\Delta d}{d}}d\theta \\ & =\frac{\varepsilon Rh}{d}\int_{\theta_{1}-\frac{\phi }{2}}^{\theta_{1}+\frac{\phi }{2}}1-\left(1-\frac{\Delta d}{d}+{\left(\frac{\Delta d}{d}\right)}^{2}\right)d\theta \\ & =\frac{\varepsilon Rh}{d}\int_{\theta_{1}-\frac{\phi }{2}}^{\theta_{1}+\frac{\phi }{2}}\frac{\Delta d}{d}+{\left(\frac{\Delta d}{d}\right)}^{2}d\theta \end{array}$$

The drive signal can be written in the form of *f = F_d_cos(ω_d_t)*, where *ω_d_* is the resonant frequency. Combining Equation (1) and the sensitivity equation, the sensitivity expression is:(32)$$\left\{\begin{array}{ll} S & =\frac{2mQ_{1}Q_{2}F_{d}}{m_{2}k_{1}w_{2}}\\ Q_{1} & =\frac{\omega_{1}m_{1}}{c_{1}}\\ Q_{2} & =\frac{\omega_{2}m_{2}}{c_{2}} \end{array}\right.$$
where *ω*_1_ and *ω*_2_ are the drive and sensing frequencies, respectively, when Δ*d* reaches its maximum value. At this time, the maximum value of vibration amplitude is reached under the sensing mode, so the relationship between the amount of capacitance change and the input angular velocity can be obtained as:(33)$$C_{rs}=\frac{\varepsilon Rh}{d}\int_{\theta_{1}-\frac{\phi }{2}}^{\theta_{1}+\frac{\phi }{2}}\frac{2m_{1}Q_{1}Q_{2}F_{d}\Omega }{m_{2}k_{1}w_{1}d}+{\left(\frac{2m_{1}Q_{1}Q_{2}F_{d}\Omega }{m_{2}k_{1}w_{1}d}\right)}^{2}d\theta$$

B. Wheel Structure

The wheel structure adopts rotating plate capacitance sensing. When the angular rate is input, the sensing mass rotates around the axis under the action of the Coriolis force, and the distance and area between it and the bottom electrode changes.

As shown in Figure 9, *θ_x_* is the deflection angle of the sensing mass when it is twisted around the Y-axis; *R*_2_ is the outer radius of the sector electrode; *R*_1_ is the inner radius of the sector electrode; *β* is the central angle corresponding to the sector electrode; *l*_1_ is the distance when the plate is parallel. In this case, the calculation formula of capacitance variation can be expressed as follows:
(34)$$\begin{array}{ll} \Delta C_{1} & =\int_{-\frac{\beta }{2}}^{\frac{\beta }{2}}\int_{R1}^{R2}\frac{\varepsilon rdrd\beta }{l_{1}-r\theta_{x}\cos\beta }-\int_{-\frac{\beta }{2}}^{\frac{\beta }{2}}\int_{R1}^{R2}\frac{\varepsilon rdrd\beta }{l_{1}+r\theta_{x}\cos\beta }\\ & =\int_{-\frac{\beta }{2}}^{\frac{\beta }{2}}\int_{R1}^{R2}\frac{\varepsilon r\left(l_{1}+r\theta_{x}\cos\beta \right)-\varepsilon r\left(l_{1}-r\theta_{x}\cos\beta \right)drd\beta }{(l_{1}-r\theta_{x}\cos\beta )\left(l_{1}+r\theta_{x}\cos\beta \right)}\\ & =\int_{-\frac{\beta }{2}}^{\frac{\beta }{2}}\int_{R1}^{R2}\frac{2\varepsilon r^{2}\theta_{x}\cos\beta drd\beta }{l_{1}{}^{2}}\\ & =\int_{-\frac{\beta }{2}}^{\frac{\beta }{2}}\frac{2\left(R_{2}^{3}-R_{1}^{3}\right)}{3l_{1}{}^{2}}\cos\beta d\beta \\ & =\frac{4\varepsilon \theta_{x}\sin\left(\frac{\beta }{2}\right)}{3l_{1}{}^{2}}\left(R_{2}^{3}-R_{1}^{3}\right) \end{array}$$

The semi-circular Y-axis detection electrode diagram is shown in Figure 10. The detection principle of the electrode is similar to that of the X-axis detection electrode. The change of the Y-axis detection capacitance value due to the influence of the Coriolis force can be obtained by the following equation:(35)$$\begin{array}{ll} \Delta C_{2} & =2\left[\int_{R_{3}-a_{1}}^{R_{3}}\frac{\varepsilon \sqrt{R_{3}^{2}-y^{2}}dy}{l_{2}-y\theta_{y}}-\int_{R_{3}-a_{1}}^{R_{3}}\frac{\varepsilon \sqrt{R_{3}^{2}-y^{2}}dy}{l_{2}+y\theta_{y}}\right]\\ & =2\int_{R_{3}-a_{1}}^{R_{3}}\frac{2\varepsilon \sqrt{R_{3}^{2}-y^{2}}y\theta_{y}dy}{\left(l_{2}-y\theta_{y}\right)\left(l_{2}+y\theta_{y}\right)}\\ & =4\int_{R_{3}-a_{1}}^{R_{3}}\frac{\varepsilon \sqrt{R_{3}^{2}-y^{2}}y\theta_{y}dy}{l_{2}^{2}}\\ & =\frac{4\varepsilon \theta_{y}{\left(R_{3}^{2}-a_{1}{}^{2}\right)}^{\frac{3}{2}}}{l_{2}^{2}} \end{array}$$
where *R*_3_ represents the outer radius of the disc, *a*_1_ represents the vertical distance between the origin and the disk-shaped electrode, *θ_y_* represents the central angle corresponding to the disc, and *l*_2_ represents the initial distance between the upper and lower plates.

The calculation principle of the sensitivity of the wheel structure is similar to Equation (32), and the calculation results are as follows:(36)$$\left\{\begin{array}{ll} S_{x} & =\frac{2\omega_{z}^{d}Q_{x}^{s}Q_{z}^{d}M_{e}}{I_{z}\sqrt{Q_{x}^{s}{\left[\left(\omega_{x}^{s}-\omega_{z}^{d}\right)\left(\omega_{x}^{s}+\omega_{z}^{d}\right)\right]}^{2}+{\left(\omega_{z}^{d}\omega_{x}^{s}\right)}^{2}}}\\ & S_{y}=\frac{2\omega_{z}^{d}Q_{y}^{s}Q_{z}^{d}M_{e}}{I_{z}\sqrt{Q_{y}^{s}{\left[\left(\omega_{y}^{s}-\omega_{z}^{d}\right)\left(\omega_{y}^{s}+\omega_{z}^{d}\right)\right]}^{2}+{\left(\omega_{z}^{d}\omega_{y}^{s}\right)}^{2}}}\\ & Q_{z}^{d}=\frac{I_{z}\omega_{z}^{d}}{C_{z}}\\ & Q_{y}^{s}=\frac{I_{1y}\omega_{1y}^{s}}{C_{1y}}\\ & Q_{x}^{s}=\frac{I_{3x}\omega_{3x}^{s}}{C_{3x}} \end{array}\right.$$
where I_z_ = $\sum_{i=1}^{3}I_{iz}$, C_z_ = $\sum_{i=1}^{3}C_{iz}$, and $\omega_{z}^{d}$, $\omega_{y}^{s}$, and $\omega_{x}^{s}$ are the X/Y-axis driving frequency, Y-axis sensing frequency, and X-axis sensing frequency, respectively. The relationship between the amount of capacitance change and the input angular velocity can be obtained as:(37)$$\left\{\begin{array}{ll} \Delta C_{1} & =\frac{4\varepsilon \sin\frac{\beta }{2}}{3l_{1}^{2}}\left(R_{2}^{3}-R_{1}^{2}\right)S_{x}\Omega_{y}\\ \Delta C_{2} & =\frac{4\varepsilon {\left(R_{3}^{2}-a_{1}^{2}\right)}^{\frac{3}{2}}}{l_{2}^{2}}S_{y}\Omega_{x} \end{array}\right.$$

## 3. Performance Analysis

### 3.1. Modal Analysis and Harmonic Response Analysis

After the gyroscope structure was determined, finite element simulation of the gyroscope structure was carried out by ANSYS. The structure was processed by single crystal silicon, and the material parameters are shown in Table 2.

The modal frequency of the gyroscope structure was obtained. The working mode of the three-axis gyroscope is shown in Figure 11. From the simulation results, it can be observed that the X/Y drive frequency was 5954.8 Hz, the Y-axis detection frequency was 5774.5 Hz, the X-axis detection frequency was 6030.5 Hz, and the maximum frequency difference was 256 Hz. The Z-axis drive frequency was 10,728 Hz, the Z-axis detection frequency was 10,725 Hz, the maximum frequency difference of Z was 3 Hz, and the frequency difference was slight, so the Z-axis sensitivity of the gyroscope was high. The variation of mechanical sensitivity with the frequency ratio is shown in Figure 12, when $\frac{\omega_{d}}{\omega_{2}}=1$, S = 0.007 μm/°/s.

Based on the modal simulation, harmonic response analysis of the structure was carried out, as shown in Figure 13. The relationship between the driving voltage and electrostatic driving force, driving voltage, and electrostatic driving torque can be obtained by taking the parameter values in Table 3 for Equations (27) and (30). The electrostatic force and torque are 0.15 μN and 6.7 × 10^−12^ N/m, respectively, as shown in Figure 14 and Figure 15, so harmonic force with amplitude of 0.1μm is applied to the gyroscope, and the displacement of the structure sensing mode is observed. The displacement of the X, Y, and Z sensing masses was 7.5, 10, and 7.7 μm, respectively.

### 3.2. Transient Shock Response Analysis

Because the gyroscope is often mounted on the gun, it will suffer a vast transient impact, which will cause the gyroscope structure to be destroyed, so the gyroscope structure often needs to have the ability to resist high overload.

The transient analysis module of ANSYS simulates the anti-high overload capability of the gyroscope. A half-sine periodic impact load of 10,000 g@5 ms was applied to the z-axis direction of the gyroscope, and the stress diagram is shown in Figure 16. At this time, the maximum stress of the structure is 300.18 MPa, which is much smaller than the allowable stress, and it can be seen that the structure has good impact resistance.

## 4. Fabrication

Accelerometers, relays, and gyroscopes are usually processed by SOG (silicon on glass) technology, which is relatively simple and has a very low parasitic capacitance [32,33,34]. The structure was manufactured with SOG. Because the aspect ratio is directly related to the gyroscope performance, the DRIE [35,36]. process was used in the etching step to ensure that the gyroscope structure has a high aspect ratio and better performance. Due to the low temperature of the anode bonding process, the influence of residual stress and strain after bonding can be weakened, and the bonding strength and stability were high. The vacuum sealing performance was good, and vacuum sealing performance is closely related to the sensitivity of the gyroscope. After synthesizing the advantages mentioned above, the anode bonding process was adopted in the processing. The processing process of the structure is shown in Figure 17.

The processing results are shown in Figure 18.

## 5. Conclusions

In this paper, a new three-axis gyroscope structure was proposed. The frame was nested by two independent systems, its design idea was given, and the motion equation of the form was deduced. The ring-shaped part of the structure is driven by static electricity with variable spacing, while the wheel structure is driven by static electricity with a variable area. During the detection, the outer resonant ring resonates along the direction of 45° and 135°, and the sensing mass of wheel will twist around the X and Y axes, respectively. The capacitance variation during the sensing mode was obtained according to the motion characteristics, and the relation equation between angular velocity and capacitance variation is obtained. In the design of the cantilever beam, S-shaped springs were used to reduce stiffness and improve sensitivity. In order to calculate the resonant frequency of the resonant ring, the stiffness calculation formula of the folded beam was given. After completing the gyroscope structure design, the specific modal frequency of the gyroscope was obtained by combining the finite element simulation software ANSYS. Moreover, the harmonic response analysis proved that the structure had high Z-axis sensitivity. In addition, in order to explore the anti-high overload performance of Z-axis direction of the gyroscope, a transient analysis was performed on it. It was found that the stress of the structure was small under the impact of 10,000 g@5 ms, which proved that the gyroscope has good performance in Z-axis anti-high overload. Finally, a structure machining method by the SOG process combined with the DRIE etching method was introduced.

## Figures and Tables

**Figure 1 sensors-22-09978-f001:**
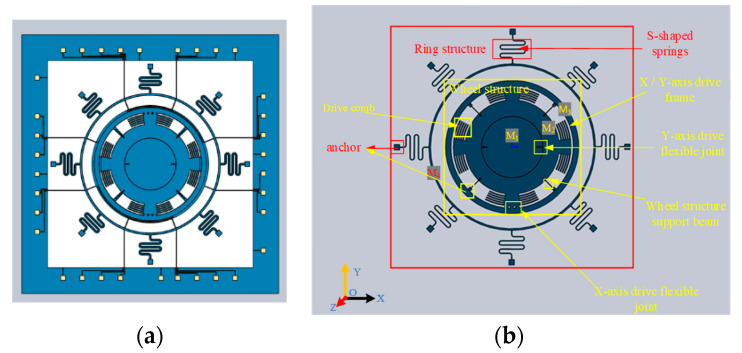
Schematic of the wheel-ring gyroscope: (**a**) Complete structure; (**b**) structure description diagram.

**Figure 2 sensors-22-09978-f002:**
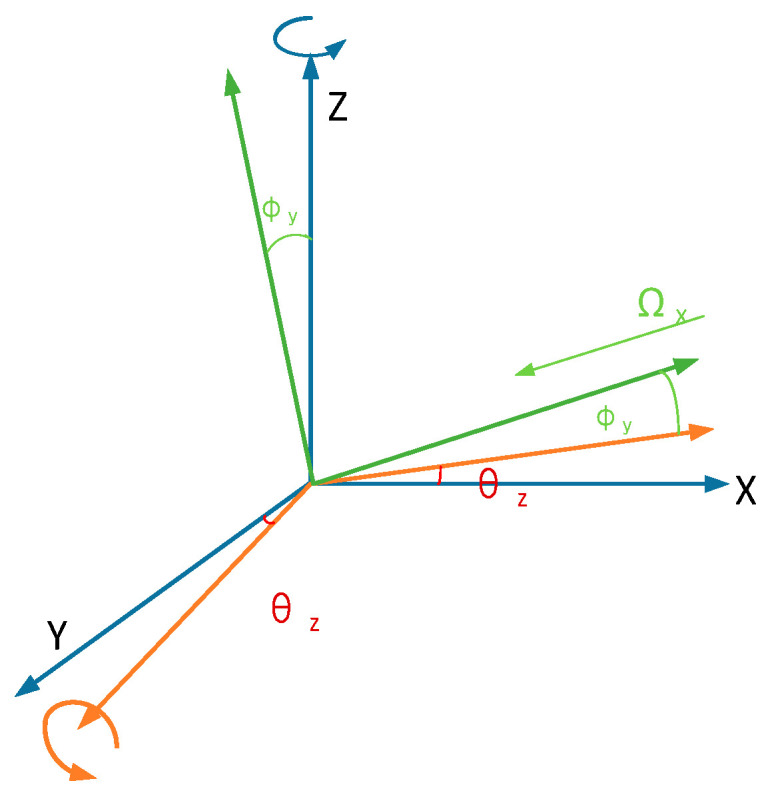
Detection the X-axis angular rate coordinate.

**Figure 3 sensors-22-09978-f003:**
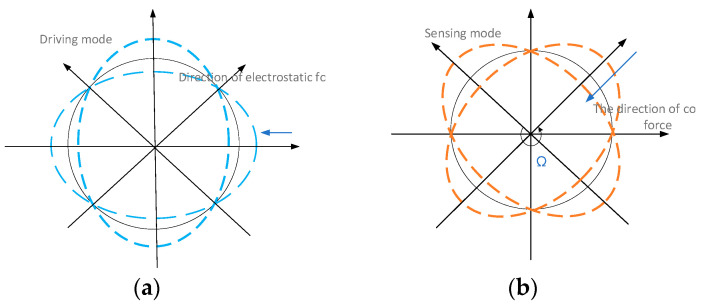
The operating modes of the resonant ring: (**a**) driving mode of the ring resonator; (**b**) sensing mode of the ring resonator.

**Figure 4 sensors-22-09978-f004:**
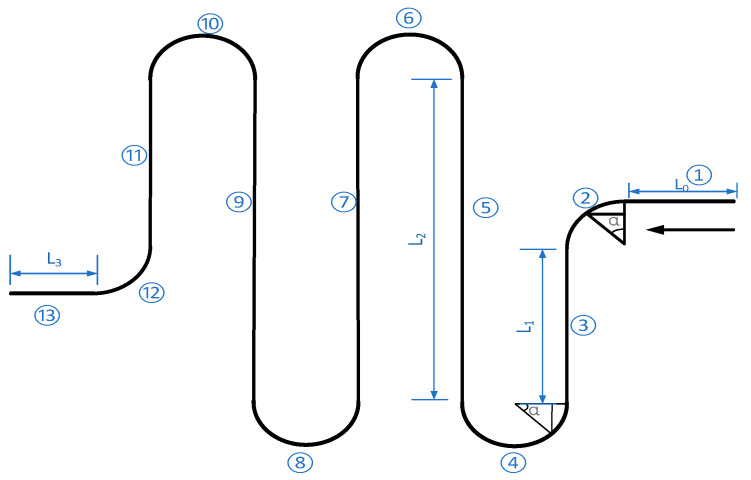
Sensitive structure of the S-shaped elastic beam’s equivalent simplified form.

**Figure 5 sensors-22-09978-f005:**
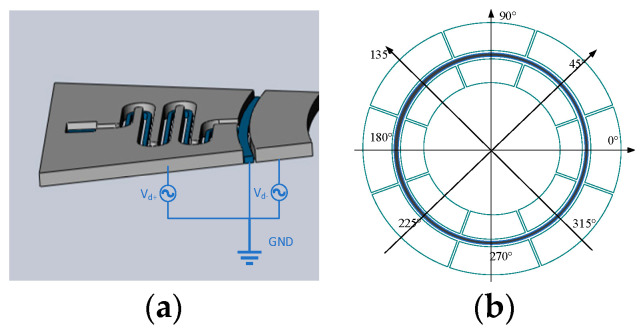
Schematic diagram of the resonant ring circuit: (**a**) The equivalent capacitance; (**b**) electrode distribution.

**Figure 6 sensors-22-09978-f006:**
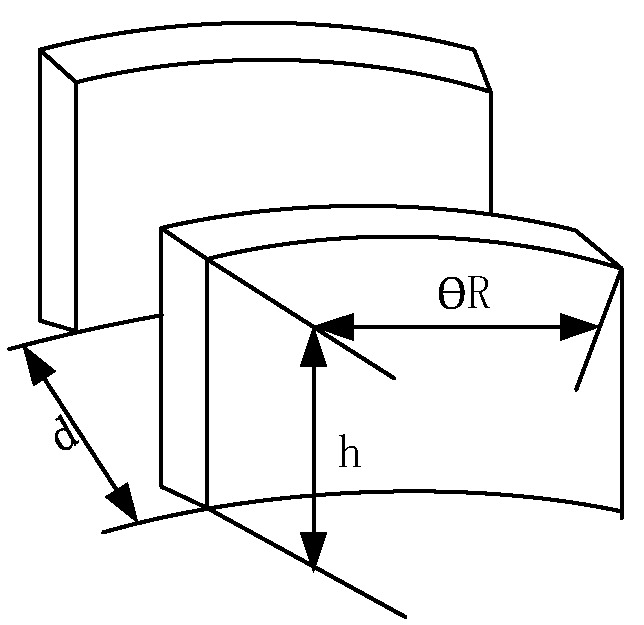
Schematic diagram of an electrostatic drive of parallel plate capacitance.

**Figure 7 sensors-22-09978-f007:**
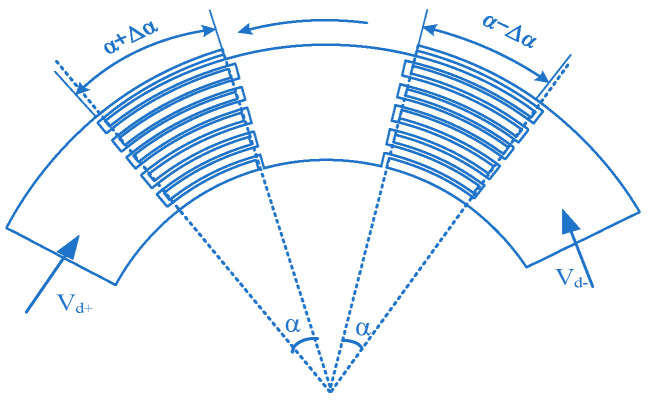
Schematic diagram of the wheel structure driving comb.

**Figure 8 sensors-22-09978-f008:**
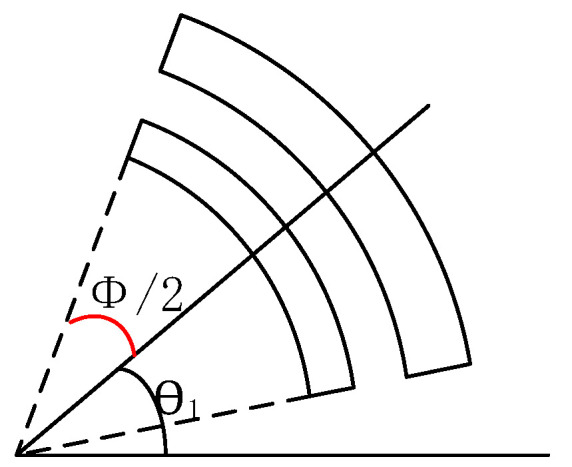
Position diagram of the resonant ring sensing electrode.

**Figure 9 sensors-22-09978-f009:**
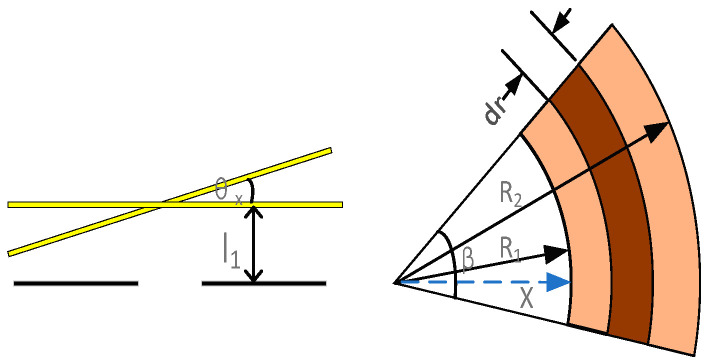
Schematic diagram of the sensing capacitance.

**Figure 10 sensors-22-09978-f010:**
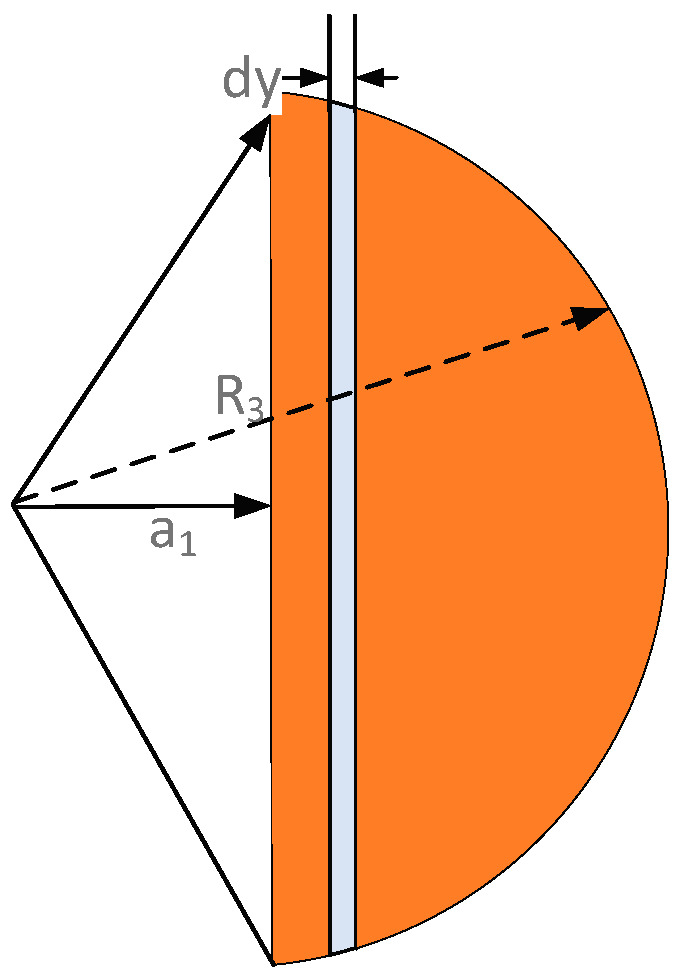
Schematic diagram of the semicircular sensing capacitance.

**Figure 11 sensors-22-09978-f011:**
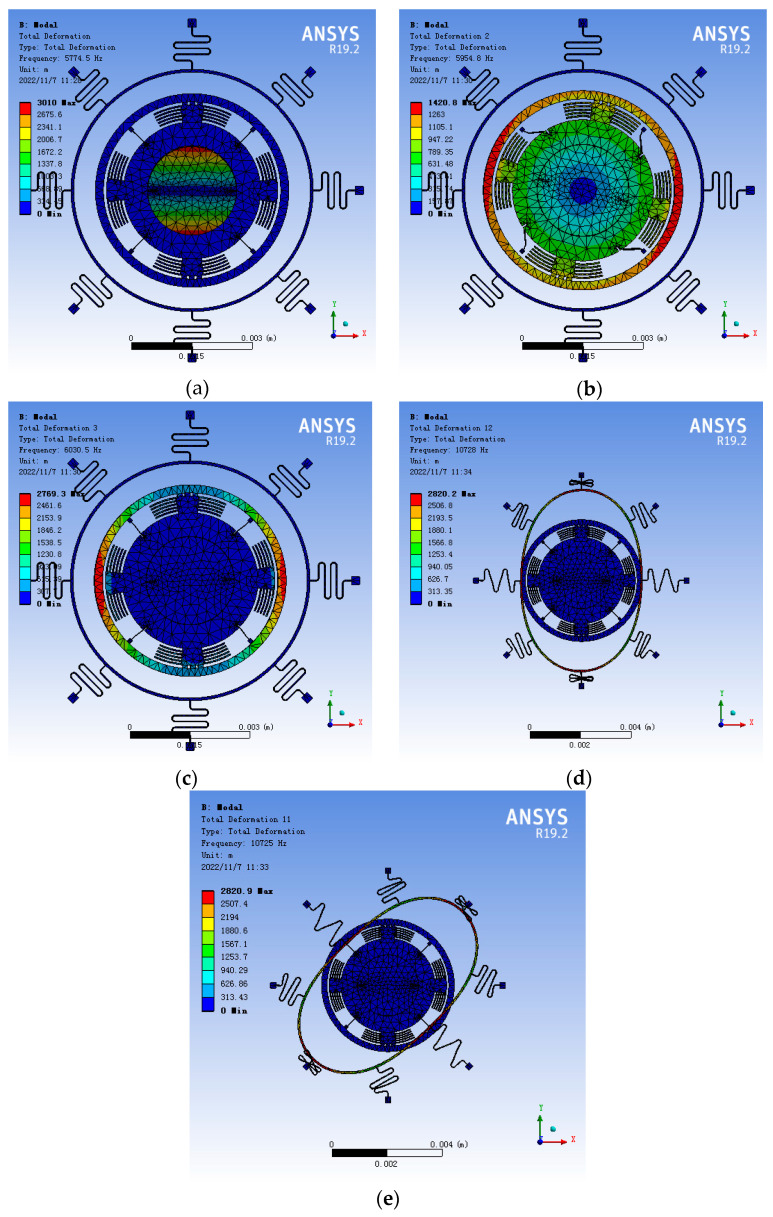
Schematic diagram of the gyroscope operating mode: (**a**) Y-axis sensing mode; (**b**) X/Y-axis driving mode; (**c**) X-axis sensing mode; (**d**) Z-axis driving mode; (**e**) Z-axis sensing mode.

**Figure 12 sensors-22-09978-f012:**
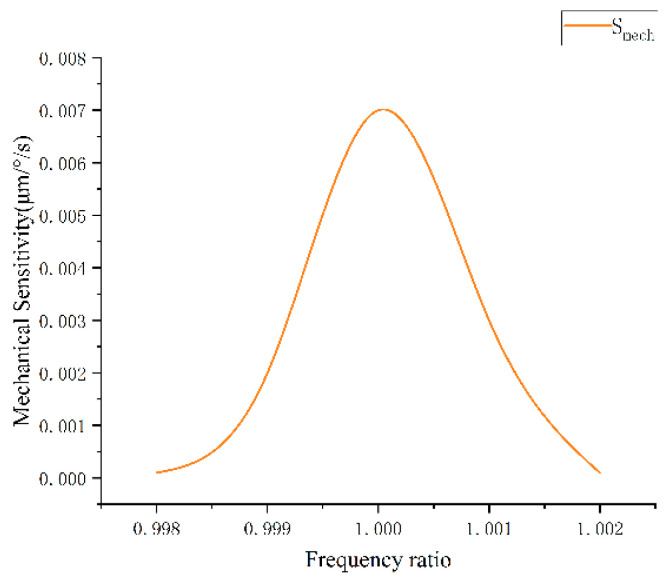
Variation of mechanical sensitivity with frequency ratio.

**Figure 13 sensors-22-09978-f013:**
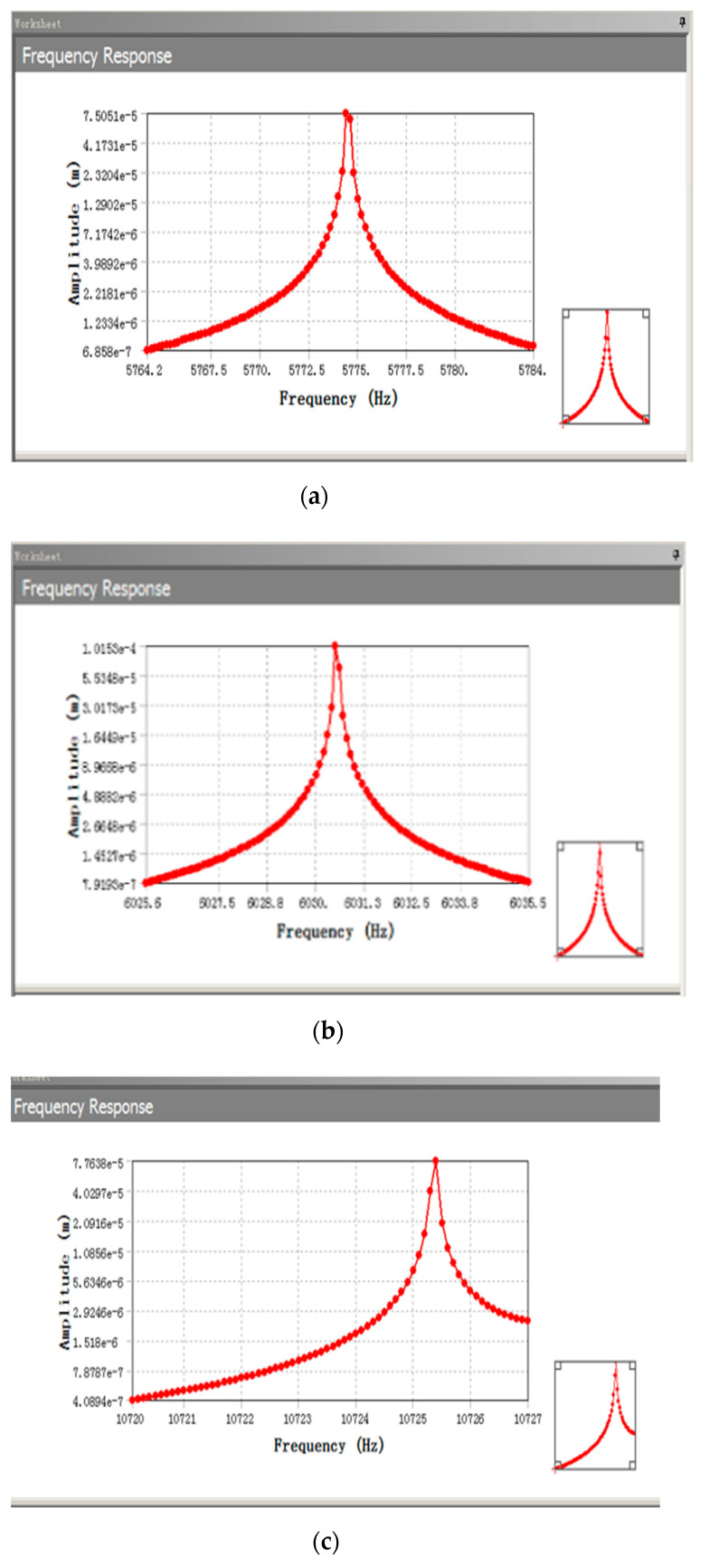
Harmonic response analysis: (**a**) X-axis sensing harmonic response; (**b**) Y-axis sensing harmonic response; (**c**) Z-axis sensing harmonic response.

**Figure 14 sensors-22-09978-f014:**
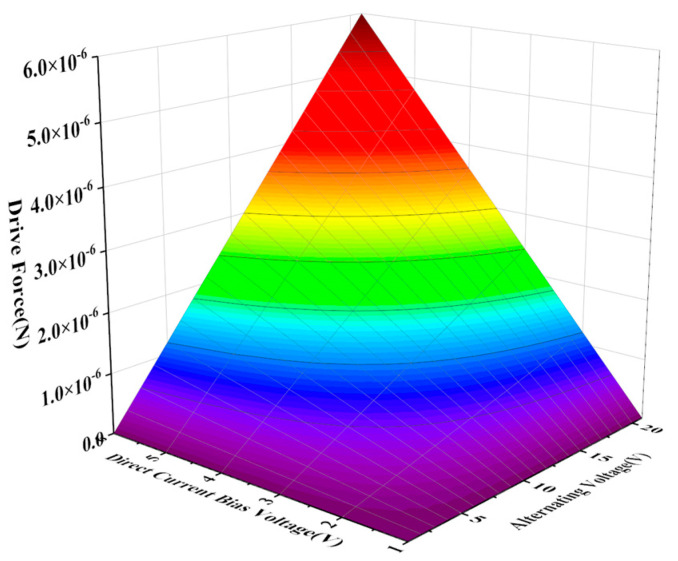
Electrostatic driving force curve.

**Figure 15 sensors-22-09978-f015:**
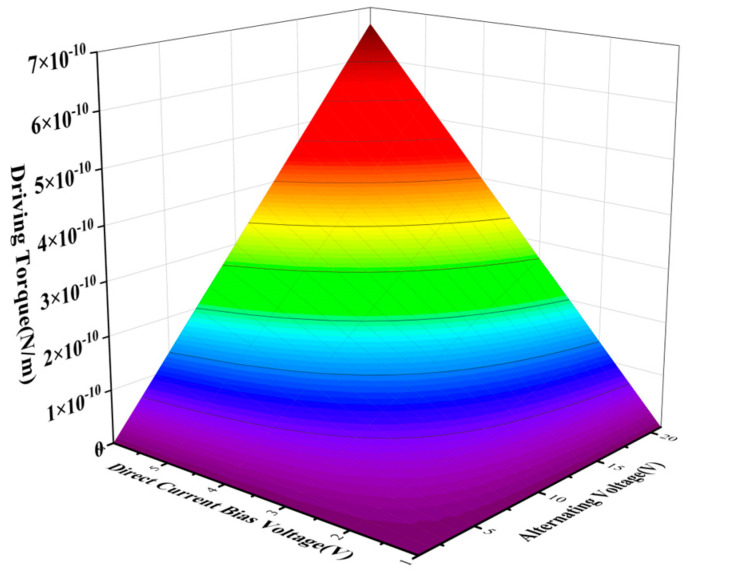
Electrostatic driving torque curve.

**Figure 16 sensors-22-09978-f016:**
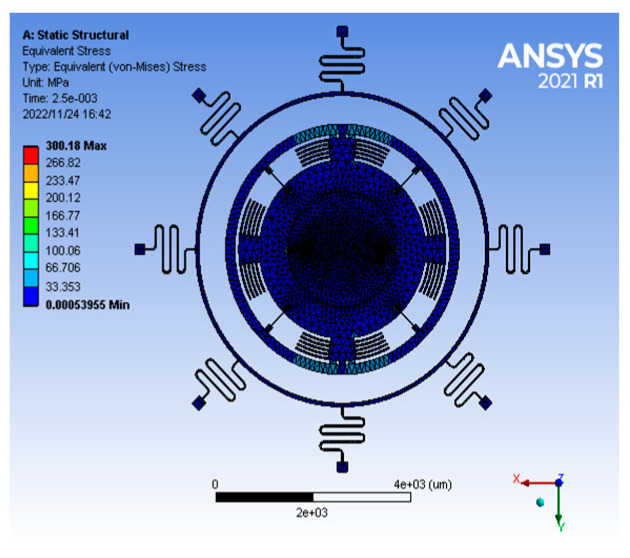
Stress analysis of gyro structure under 10,000 g impact.

**Figure 17 sensors-22-09978-f017:**
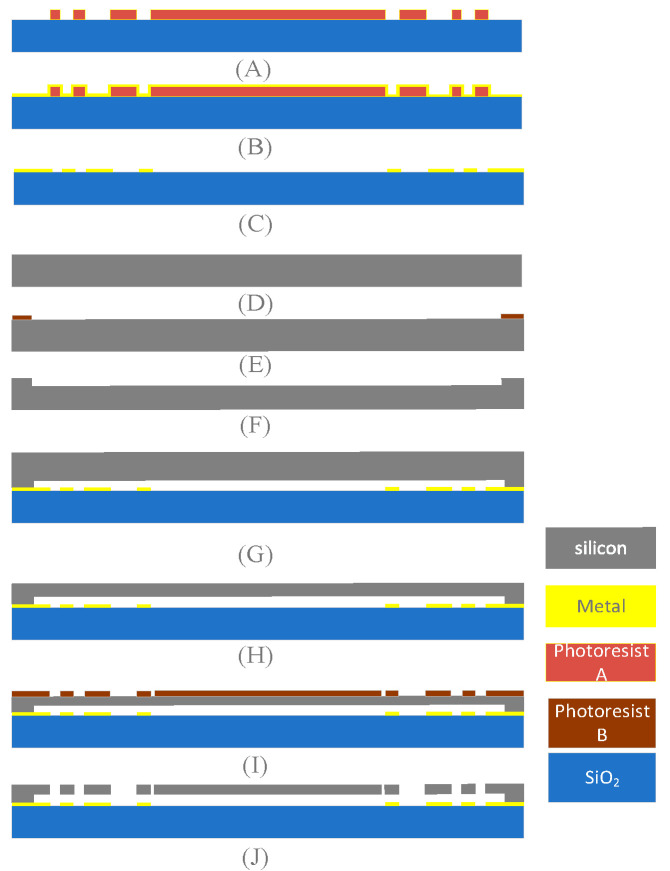
The fabrication process of the TAG. (**A**) After cleaning and drying the glass wafer, the glass substrate was spin-coated with AZ4620 photoresist and baked. After baking, the mask was aligned with the glass substrate and irradiated with a UV light source to pattern the glass substrate. (**B**) The electrical connection layer was formed by sputtering Cr/Au on the glass substrate to a thickness of 200 nm. (**C**) Photoresist was removed with an organic solvent (usually ethanol and acetone). (**D**,**E**) A 4-inch, 400-μm-thick <111> silicon wafer was used, and AZ4620 was coated and patterned on the back side of the wafer. (**F**) DRIE was used to etch a 2 μm depth for preparing the anchor of the gyroscope. The residual photoresist was removed at the end of the etching process. (**G**) The silicon and metallized glass wafers were anodically bonded on an EVG510 bonding machine with a bonding temperature of 400 °C and a bonding voltage of 800 V. (**H**) After step G, the silicon wafer thickness was thinned to 100 μm. (**I**) Photoresist was applied on the front side of the silicon wafer and a second etching is performed with an etching depth of 100 μm. (**J**) Finally, etching was completed and the structure is released.

**Figure 18 sensors-22-09978-f018:**
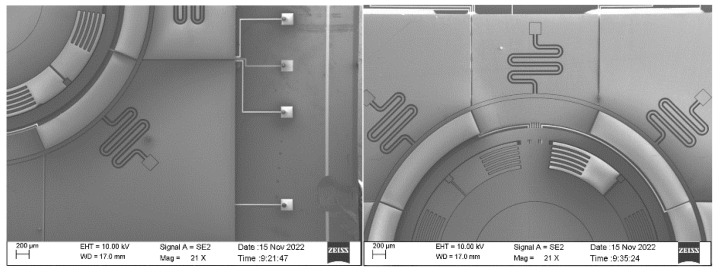
Image of TAG under the microscope.

**Table 1 sensors-22-09978-t001:** Comparison with a previously reported 3-axis gyroscope.

Institutions	Size (mm × mm)	Z-Axis Shock Resistance	Year
SEU [1]	9.80 × 9.80	Unknown	2015
FSL [7]	2.90 × 2.25	Unknown	2017
KEIO [8]	5.00 × 5.00	Unknown	2020
UCAS [9]	3.57 × 3.47	Unknown	2014
UCD [10]	3.20 × 3.20	Unknown	2015

**Table 2 sensors-22-09978-t002:** The material parameters of gyroscope.

Property	Value
Density	2330 kg/m^3^
Young’s Modulus	169 GPa
Poisson’s Ratio	0.27

**Table 3 sensors-22-09978-t003:** Values for design parameters.

Design Parameter	Value
Ring Radius, *R*	3000 μm
Central Angle, *θ*	40°
Thickness, *h*	80 μm
Plate Spacing, *d*	5 μm
Overlapping Angle, *α*	18°
Comb Gap, *h_d_*	5 μm
Moving Comb Radius, *r_i_*	(1665 + 80i) μm (i = 1,2..,6)
Direct Current Bias Voltage, *V_D_*	5 V
Quality Factor Q	3000

## Data Availability

Not applicable.

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
