# Peer review of "Design and Fabrication of a Novel Wheel-Ring Triaxial Gyroscope"

_sensors, 2022, doi:10.3390/s22249978_

Round 1

Reviewer 1 Report

This work could represent a thorough study of a novel gyroscope based on a new triaxial ring silicon structure. Indeed, this work could be the basis for an interesting article. However, it requires a careful revision of the explanations and a comparison with the state of the art. The author could polish up the work in terms of English

1.The novelty and research gaps could be emphasized more at the end of the introduction. First of all, a lot of work has already been published on the gyroscope. Also, why is your methodology unique and why have previous studies not addressed these details?

2. A comparison with the state of the art needs to be made in a discussion section (dimension, structure, frequency, factor quality...). Please focus on the electrotractive drive.

4. Everything in this section about experiments is known, so why the interest? it is complicated to understand this section

5. we notice some inconsistencies, like gyro. Please change gyro to gyroscope.

6. The labeling of figures 11 and 12 is a mistake. Please correct it.

7.The English language is clearly not proofread; there are syntax and spelling errors in every other paragraph. Sometimes these errors make it difficult to understand the meaning of the sentence

8. several papers need to be cited, especially the most recent ones. At this stage, this work is not very convincing. Please make an effort to provide bibliographic references to more recent results.

9. Put the scale bar in figure 14 figures. It would also be more interesting to include a SEM image.

10. Apart from the structured design, it is very difficult to understand the logic of your study. It is not a rigorous work and that is a pity. The results are presented in a concise manner. This paper should be rewritten and take more time It is difficult to read this paper. Some sections need to be moved and inserted elsewhere to allow for better sequencing of explanations.

11. I suggest rewriting this paper with specific sections such as experimental setup, results, discussion, and conclusion.

12. What are the experimental etching conditions and what tools did you use? Recent work on DRIE silicon etching has recently been published in Journal of Vacuum Science & Technology B 37, 021206 (2019); https://doi.org/10.1116/1.5081503, and Phys. Status Solidi A, 216: 1900324. https://doi.org/10.1002/pssa.201900324, for reference and comparison of your high aspect ratio engraving technique. And relate the originality of your etching technique to these works.

13. EVG parameters need to be clarified and explain.

14 This work needs to be better analyzed, the quality factor measured in air and in vacuum, and better commented on.

 15. At this time, I unfortunately cannot recommend this work.

Author Response

Dear Expert:

Thank you for taking the time to read this article and giving specific and inspiring suggestions. We have revised this article after carefully reading your comments.

Reviewer 2 Report

The paper presents a novel design for a wheel ring triaxial gyro.

2. The design is innovative and shows a good understanding of the principles of gyroscopes.

3. The paper is well written and easy to follow.

4. The fabrication process is clearly described and the results are presented in a clear and concise manner.

5. The paper includes a discussion of the results and their implications.

6. The paper is well referenced and includes a comprehensive list of relevant literature.

7. The paper is of a high standard and would be suitable for publication in a peer-reviewed journal.

8. The paper includes a detailed discussion of the design and implementation of the new method.

9. The paper is of a high standard and would be suitable for publication.

Author Response

(The authors gave the same response as above.)

Reviewer 3 Report

In the paper, the authors have presented a novel design of a MEMS gyroscope. Below are some of my comments;

1.     There are multiple formatting errors, English grammar and sentence structuring errors that must be improved.

2.     In the Introduction section, authors should include a discussion for the need of a new ring type MEMS gyroscope structure. What are current limitations of the existing MEMS gyroscopes that the proposed design can overcome?

3.     Figure 1(b) should be enough to explain the proposed MEMS gyroscope design. Moreover, the labeling of the MEMS gyroscope should be clear.

4.     Some of the equations (Eq. 7, 12, 13 etc) seems to be included as picture which should be properly written.

5.     In the FEM analysis, authors should mention the material properties used.

6.     The harmonic response of the MEMS gyroscope is strongly dependent on the air damping. Authors must perform air damping analysis first and then include the damping coefficients in the FEM analysis to analyze the frequency response.

7.     Authors must mention the actuation force estimation and corresponding values used in the FEM based harmonic analysis.

8.     How the drive and sense mode resonance frequency matching will be achieved in the presence of microfabrication process uncertainties and environmental variations like temperature and pressure.

9. Authors must include a comparison of the results obtained through the FEM simulations and developed mathematical model. 

Author Response

(The authors gave the same response as above.)

Round 2

Reviewer 1 Report

I believe this paper will be of particular interest for the readership of your journal.

Author Response

Thank you!

Reviewer 3 Report

Authors have mentioned the main contribution of this work in the introduction section as "The design has a tiny size, simple structure, excellent resistance to overload and high sensitivity." I have following main doubts;

1. There is no comparison of the proposed gyroscope design size in comparison to that already presented in the literature.

2. How authors define simple structure? As per my view this is simple structure in comparison to the vibratory MEMS gyroscope presented in the literature owing to the fact that there multiple joints, mechanical springs and parallel plate actuation is being used. 

3. For the "excellent resistance to the overload" claim, there must be some shock resistance analysis to prove this contribution and comparison of shock resistance in comparison to the literature.

4.  Again, how authors have claimed the high sensitivity while there is no discussion on the sensitivity  in the manuscript. Authors must have included experimental or simulation based results for the sensitivity of the gyroscope in the presence of input rotation and compare it with the gyroscope designs presented in the literature.

In the present, form I am afraid that this manuscript can not be accepted since authors have not provided sufficient results to justify their claims.  

Author Response

Thank you!
